# The etiology of attention deficit disorder with hyperactivity: A protocol for an umbrella review

**Wenhao Su, Hairong Jia, Luo Yang, Jiaqi Zhang, Zhaoyang Wei, Pepertual Tsikwa, Yanru Wang** *

School of Nursing, Zhejiang Chinese Medical University, Hangzhou, Zhejiang, China

* Wangyanru001@outlook.com

## Abstract

### Introduction

Attention deficit hyperactivity disorder (ADHD) is one of the common neurodevelopmental disorders and is widely prevalent worldwide. The primary symptoms of ADHD include inattention, impulsivity, and hyperactivity, which significantly impact the cognitive, behavioral, and emotional dimensions of individuals. These disorders often continue throughout adulthood and, along with associated complications, affect various domains such as personal health, academic achievement, and social interactions. The pathogenesis and contributing causes of ADHD remain unclear at present. Therefore, this study aims to perform an umbrella review of systematic reviews and meta-analyses (SRMAs) to systematically assess the quality of methodologies, potential biases, and validity of all epidemiological evidence related to risk factors for ADHD while offering a comprehensive summary of the evidence regarding these risk factors.

### Methods and analysis

This study follows the Preferred Reporting Items for Systematic Review and Meta-Analysis Protocols (PRISMA-P) and Cochrane Handbook. We will systematically search 6 databases, including The Cochrane Library Central, PubMed, Embase, Web of Science, CINAHL, and Scopus from the initial period up until 2024 (last update). We will assess the quality of the included SRMAs using the tool to assess risk of bias in systematic reviews (ROBIS), the methodological quality of systematic reviews (AMSTAR)-2, PRISMA-2020, and the grade of recommendations assessment, development and evaluation (GRADE). Two authors will use the ecological models of health behavior to classify the causes and risk factors of ADHD. Finally, we will provide descriptive and comprehensive recommendations for clinical practice and future research.

### Trial registration

PROSPERO (CRD42024597126).

**Data Availability Statement:** No datasets were generated or analysed during the current study. All relevant data from this study will be made available upon study completion.

**Funding:** The author(s) received no specific funding for this work.

**Competing interests:** The authors have declared that no competing interests exist.

**Abbreviations:** SRMAs, Systematic Reviews and Meta-Analyses; PRISMA-P, Preferred Reporting Items for Systematic Review and Meta-Analysis Protocols; PRISMA 2020, The Preferred Reporting Items for Systematic Reviews and Meta-Analysis 2020; ROBIS, Risk of Bias in Systematic Reviews; AMSTAR-2, A Measurement Tool to Assess Systematic Reviews-2; GRADE, Grade of Recommendations Assessment, Development and Evaluation; PECOS, Population, Exposure, Comparator, Outcome, Study Designs, Setting; DSM-5, Diagnostic and Statistical Manual of Mental Disorders, Fifth Edition; ASRS, ADHD Self-Report Scale.

# Introduction

One of the most prevalent neurodevelopmental disorders is attention deficit hyperactivity disorder (ADHD), which affects 7.2% of children worldwide [1], with symptoms that can last from 40% to 60% of a child's life [2,3], and a prevalence of 2.58% among adults [4]. ADHD shows either impulsive hyperactivity, or attention deficit disorder, even both of them. Learning difficulties [5], as well as neurological conditions such as Tourette's syndrome, autism, and anxiety [6–8], are more common in children diagnosed with ADHD. As adults, the patient's condition would be more terrible, adult ADHD patients are more likely to live through substance misuse, depression, bipolar illness, personality disorders, schizophrenia [9–12], etc.

Children with ADHD experience significant deficiencies in learning and social skills, which may lead to enduring challenges in personality development, interpersonal communication, and social adaptation. It used to be commonly accepted that the primary causes of ADHD development were genetics, alterations in the social environment, exposure to toxins, and parental education. But more research in the past few years has revealed that hypothyroidism [13], trauma [14], cerebellar dysfunction [15], and gut microbiome dysbiosis [16] are among the other possible pathogenic causes [17] for ADHD. Consequently, identifying the causes of ADHD is essential for its management and prevention, offering other potentially useful strategies for treatment.

Umbrella review is a high-level review that uses analysis of systematic reviews and meta-analyses (SRMAs) to synthesize a specific issue and provide references for clinical practice and future research initiatives [18]. At present, no umbrella review related to ADHD has been published. Therefore, our study aims to systematically assess the quality of methodologies, potential biases, and validity of all epidemiological evidence related to risk factors for ADHD while offering a comprehensive summary of the evidence concerning risk factors.

# Methods

This protocol complies with the Preferred Reporting Items for Systematic Review and Meta-Analysis Protocols (PRISMA-P) guidelines [19]. We have finished the PRISMA-P document (S1 File). This umbrella review will also be under the methodological guidance of the Joanna Briggs Institute Manual for Evidence Synthesis of Umbrella Reviews [20], and the Cochrane Handbook for conducting systematic reviews [21]. We will also search grey literature to find more potential good studies. We have created the search strategy for all databases (S2 File). This protocol was registered on PROSPERO (CRD42024597126).

## Ethics and dissemination

Ethical approval is not required for this umbrella review. We will seek to submit the results for publication in a peer-reviewed journal or present it at conferences.

## Eligibility criteria

The eligibility criteria were established based on the Population, Exposure, Comparator, Outcome, Study Designs, Setting (PECOS) statement [22]. The PECOS statement is shown in S1 Fig.

## Population

This review aims to examine the risk factors and etiology of ADHD through SRMAs. SRMAs must involve participants with ADHD or animal experimental models that simulate ADHD. This methodology will involve SRMAs of observational study designs (cohorts, cross-sectional,

case-crossover, and time series) as well as experimental study designs (RCTs). Other research not relevant to ADHD etiology or risk factor analysis will be omitted.

## Exposure

Any factors which might influence the development or etiology of ADHD. Participants influenced by sociopsychological variables, environmental toxicant exposure, genetic biomarkers, and other risk factors or causes will be considered. We will include any SRMAs with at least one clearly established risk factor for ADHD, such as demographic, disease-related, or psychosocial factors. Risk variables could be provided with or without adjusted effect sizes across categories, such as odds ratios (ORs), relative risks (RRs), and hazard ratios (HR) with 95% confidence intervals (CIs).

## Comparator

ADHD patients/animals who have not been exposed to the risk factors under investigation or trial, such as neurotypical children and animal models.

## Outcome

SRMAs take ADHD etiological factors or risk factors as outcomes will be considered. We will take the Diagnostic and Statistical Manual of Mental Disorders, Fifth Edition (DSM-5) definition of ADHD as the criteria. For example, adult patients have a diagnosis of ADHD and score$\geq$14 on the ADHD Self-Report Scale screener items (ASRS), and children patients diagnosed with ADHD by doctors with DSM-5.

## Exclusion criteria

To maximize the potential studies, no exclusions will be applied based on the presence of comorbidities. If possible, we will classify the included comorbidities and conduct subgroup analysis based on the type of comorbidities.
   (1) Any study design except SRMAs;
   (2) Studies that did not report at least one possible cause for ADHD etiology;
   (3) Studies are written in other languages than English;

## Study design

To be included, SRMAs need to focus on the question about risk factors or causes of ADHD, and clearly describe an explicit and reproducible methodology, including systematic searching strategies, systematic selection of included studies, predefined eligibility criteria, critical quality appraisal of included studies, and quantitative or qualitative synthesis of results. Systematic reviews can include studies with prospective/retrospective cohort design, analytical cross-sectional design, case-control design, and randomized controlled trials. Protocols will not be considered. We will contact the first author when we get the abstract only, then we will send 2 emails twice a week on Monday and Friday for 2 months. If the author does not give us the full text, we will exclude it.

## Study selection

The software Endnote X9 will be used to import all search results and perform deduplication. Then the deduplicated literature will be imported into Rayyan for screening. The titles and abstracts will be examined by three independent reviewers to see if they satisfy the requirements for inclusion. In the event of a disagreement, the three reviewers will discuss and make

a decision together. The selected study texts will be fully read by another two impartial reviewers to see if they satisfy the inclusion requirements. In the event of a disagreement, the third reviewer will decide in the end. If we are unable to obtain the full text, we will send emails to the author every Monday and Friday within 2 months to obtain the full text, if we have not received the full text, that study will be deleted from inclusion.

## Overlapping assessment

We will use the corrected covered area (CCA) for the assessment of overlap. We will compute CCA by constructing a graphical cross-tabulation (citation matrix) and assess the overlap rate of the original articles used in the SRMAs studies. The extent of overlap is categorized into four levels: extremely high overlap ($\geq$15%), high overlap (11%~15%), moderate overlap (6% to 10%), and low overlap ($\leq$ 5%). The formula for calculating CCA is as follows: CCA = N—r / [(r × c)—r]. "N" denotes the number of repeated investigations, "r" signifies the count of original studies, and "c" indicates the number of SRMAs. If the CCA is less than 10%, we deem the overlap acceptable and will proceed with direct merging for analysis. If CCA$\geq$10%, we will prioritize selecting studies that (1) are published in the Cochrane Library, (2) are the most recent and relevant to our issue, (3) involve a larger number of participants, and (4) exhibit high quality as assessed by A Measurement Tool to Assess Systematic Reviews-2 (AMSTAR-2) and Grade of Recommendations Assessment, Development and Evaluation (GRADE) [23].

## Data extraction

Two reviewers will independently extract data. Such as author of the study, year of publication, country of origin, kind of study (human/animal, cross-sectional, cohort, case-control, randomized controlled trial), characteristics of participants, number of cases and total participants, diagnostic criteria for ADHD, estimates of relative risk (RR), odds ratio (OR), or hazard ratio (HR) with 95% confidence intervals (CIs); Three independent reviewers will extract the quality assessment tool for heterogeneity estimation ($I^2$, Cochrane's Q test), Egger's test, Begg's test, funnel plot, risk variables, and results/findings utilizing Excel 2023 software. If there is a disagreement, the final decision will be made by the third reviewer.

We will employ the $I^2$ statistic and Cochrane Q test for heterogeneity analysis, along with sensitivity analysis (utilizing the trim and fill approach and exclusively using a specific study type, such as prospective studies) to evaluate the stability of research findings. We will utilize the OR as a consolidated effect measure for conversion, as indicated in Paolo's report (S3 File) [24]. We will utilize the quality assessment report via PRISMA-2020 to evaluate the robustness of the study findings. We will examine the author's conflict of interest statement for any possible conflicts of interest.

## Quality assessment

**(1) Risk of bias assessment.** The risk of bias in systematic reviews (ROBIS) tool will be used to assess the risk of bias present in the SRMAs by two reviewers [25]. If there is a disagreement, the final decision will be made by the third reviewer. ROBIS includes three aspects: ①assessing relevance (selected according to the situation); ②Determine the level of bias risk in the process of developing system evaluations; ③Assess the risk of bias in systematic review evaluations. Each aspect of the ROBIS tool contains several landmark questions, and the answers to these questions are "yes", "possible", "no", and "possible or no information". The risk of bias in the final system evaluation is judged as low risk, high risk, or unclear. Two reviewers will independently evaluate the risk of bias quality of all included research. Should there be no agreement, the two reviewers will meet to make the final decision. In the research

findings, the outcomes of the risk of bias will also be reported. When over 10 studies are included, we will use Egger's for publication bias analysis (P<0.1 indicates the statistical significance of publication bias) [26].

**(2) Assessment of methodological quality.** Two reviewers will independently use AMSTAR-2 to evaluate the methodological quality of the included SRMAs. This tool consists of 16 items, covering key issues included in the research, proposal, literature search, literature screening, data extraction, basic characteristics of the original research included, data analysis, and conflicts of interest, among others, including 7 key areas [27]. By evaluating the compliance of research with each criterion item, the overall quality of the system evaluation can be determined [12], which can be classified as "high" (no more than one noncritical area has defects: the system evaluation provides an accurate and comprehensive summary of outcomes); 'Medium' (defects in more than one noncritical area: systematic evaluation provides a more accurate and comprehensive summary of outcomes); Low "(with or without defects in one key area and non-key areas: the system evaluation cannot provide an accurate and comprehensive summary of outcomes); Or "extremely low" (with defects in more than one important area with or without non-important areas: the system evaluation cannot provide an accurate and comprehensive summary of outcomes based on the obtained data) at four levels.

**(3) Assessment of report quality.** The preferred reporting items for systematic reviews and meta-analysis 2020 (PRISMA 2020) scale will be used to evaluate the quality of SRMAs that met the inclusion criteria [28] by 2 reviewers. If there is a disagreement, the final decision will be made by the third reviewer. The PRISMA 2020 reporting standard consists of seven parts: title, abstract, introduction, methods, results, discussion, and other information, with a total of 27 items (42 sub-items). According to the completeness of the report, each item will be evaluated as a complete report (Y) with a score of 1, partial reports (PY) with a score of 0.5, and unreported (N) with a score of 0, for a total score of 42. A report with a completeness of over 80% (33–42 points) is considered "relatively complete" and rated as high-quality; A report with a completeness of over 60% (25–32 points) is considered to have some defects and is rated as of moderate quality; A report with a completeness of less than 60% (<25 points) is considered to have "relatively serious information gaps" [28,29].

**(4) Assessment of evidence quality.** Two reviewers independently used the GRADE tool to evaluate the credibility of evidence in meta and systematic reviews. If there is a disagreement, the final decision will be made by the third reviewer. The GRADE system judges the increase or decrease in evidence confidence based on five factors that may lower the quality level (risk of bias, inconsistency, indirectness, inaccuracy, and publication bias) and three factors that may increase the quality level (dose-response relationship, large effect size, and negative bias evaluation of the confidence level of each outcome evidence). Ultimately, the evidence will be classified into four levels: high, moderate, low-level, and extremely low-level. And divide the epidemiological evidence for each risk factor into four categories (evidence stratification) [23]:

- ·Class I (convincing evidence): >1000 cases (or >20,000 participants for continuous outcomes), statistical significance at $p<10^{-6}$ (random effects), no evidence of small-study effects and excess significance bias; 95% prediction interval excluded the null, no large heterogeneity ($I^2<50\%$);

- ·Class II (highly suggestive evidence): >1,000 cases (or >20,000 participants for continuous outcomes), statistical significance at $p<10^{-6}$ (random effects) and largest study with 95%CI excluding the null value;

- ·Class III (suggestive evidence): >1000 cases (or >20,000 participants for continuous outcomes) and statistical significance at $p<0.001$;

- ·Class IV (weak evidence): The remaining significant associations with $p<0.05$;·NS: non-significant: $P>0.05$.

For risk factors classified as Class I and Class II, sensitivity analysis will be conducted when they can have a significant impact on the overall results.

## Data summary

We will use the ecological models of health behavior to classify the risk factors of ADHD. According to this model, risk factors can be divided into five aspects [30]: individual innate characteristics (including age, gender, race, height, weight, BMI, medical history, and susceptible genes), behavioral lifestyle (including diet, exercise, smoking, alcohol, hobbies, etc.), interpersonal network (including marital status, number of family members, number of friends, social relationships, etc.), socioeconomic status (including occupation, personal income level, family economic level, education level, debt, etc.), macro environment (including economy, urban or rural environment, policy, etc.). We have created the operation guideline of these 5 aspects (S4 File). Based on the model operation guidelines, two reviewers will classify the influencing factors. If there is a disagreement, the third reviewer will make the final decision. we will establish a classification list of ecological models of health behavior in advance, clarify the factors under each category, and make it easier to ensure the accuracy of classification when extracting data. This could help in identifying similarities and variations among significant study-included factors. We will give a detailed explanation of the outcomes of the included systematic reviews, explaining the study characteristics in relation to the intended study population, the quantity and variety of studies included, and the conclusions reached in light of the variables influencing ADHD. The differences and similarities between the included research results will be the main topic of discussion. We will use descriptive and narrative synthesis to summarize and refer to the quality of evidence.

## Analysis of subgroups or subsets

The extracted data will be tabulated to help identify commonalities and variations in relevant parameters across the included research. The tabulation will also assist in identifying subgroups within the dataset. We will lay out groupings of similar systematic reviews and/or outcome measures to categorize similar populations or outcomes.

## Discussion

ADHD is not only more prevalent in children, but it also persists into adulthood, significantly detrimental to a patient's social, psychological, and physical abilities. Nevertheless, the precise etiology and pathophysiology of ADHD remain poorly comprehended. In addition, the effectiveness of medication and non-pharmacological treatments for ADHD is very weak [31,32] because of the complexity of its underlying causes, imposing a significant burden on patients, families, and healthcare systems [33].

To get further insight into the causes of ADHD, ranging from micro genes and pathogenesis to macro social life and environmental pollutants, we will examine SRMAs of risk variables correlated to the disorder. We will induce the ecological models of the health behavior and evaluate the quality of the included studies from 4 aspects: risk of bias, methodology, report quality, and quality of evidence.

Currently, some umbrella reviews on neurodevelopmental disorders are being conducted, including an analysis of the etiology of autism spectrum disorder [34], the relationship between analogic drug use in pregnancy and neurodevelopmental disorders [35], and the prevention of comorbid psychiatric disorders among people with autism spectrum disorder [36]. However, no such study has been undertaken into the causes of ADHD.

There are some challenges in this article as well. Firstly, we limit our search to English-language studies, which may leave some potentially important research that has been published in other languages. Secondly, we will also search grey literature, it helps us to get more related studies, but it may also induce some bias.

Despite certain limitations, this is currently the first Umbrella review to investigate the etiology of ADHD. This umbrella review will assess the benefits and drawbacks of current evidence-based data derived from SRMAs regarding the risk factors associated with ADHD. It aims to enhance understanding of the potential risk factors influencing the onset and progression of ADHD from various perspectives, establish a theoretical foundation for the formulation of more clinically effective prevention and intervention strategies for ADHD, and offer guidance for future clinical research endeavors.

## Supporting information

**S1 Fig. PECOS statement.**
(TIF)

**S1 File. PRISMA-P 2015 checklist.**
(PDF)

**S2 File. Search strategy.**
(PDF)

**S3 File. Possible conversions of some effect sizes to equivalent ORs.**
(PDF)

**S4 File. The operation guideline for the ecological models of health behavior.**
(PDF)

## Author Contributions

**Conceptualization:** Wenhao Su.

**Data curation:** Luo Yang, Jiaqi Zhang.

**Formal analysis:** Wenhao Su, Hairong Jia, Jiaqi Zhang.

**Investigation:** Yanru Wang.

**Methodology:** Wenhao Su, Luo Yang.

**Supervision:** Hairong Jia, Yanru Wang.

**Writing – original draft:** Wenhao Su, Zhaoyang Wei, Pepertual Tsikwa.

**Writing – review & editing:** Wenhao Su, Yanru Wang.

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
