## [Decision Letter · Decision Letter 0]

3 Dec 2024

PONE-D-24-44751The Etiology of Attention Deficit Disorder with Hyperactivity: A systematic review protocolPLOS ONE

Dear Dr. Yanru,

Thank you for submitting your manuscript to PLOS ONE. After careful consideration, we feel that it has merit but does not fully meet PLOS ONE’s publication criteria as it currently stands. Therefore, we invite you to submit a revised version of the manuscript that addresses the points raised during the review process.

We look forward to receiving your revised manuscript.

Kind regards,

Carmen Concerto

Academic Editor

PLOS ONE

Journal Requirements:

Reviewers' comments:

Reviewer's Responses to Questions

**Comments to the Author**

1. Does the manuscript provide a valid rationale for the proposed study, with clearly identified and justified research questions?

Reviewer #1: Partly

Reviewer #2: Yes

2. Is the protocol technically sound and planned in a manner that will lead to a meaningful outcome and allow testing the stated hypotheses?

Reviewer #1: Yes

Reviewer #2: Yes

3. Is the methodology feasible and described in sufficient detail to allow the work to be replicable?

Reviewer #1: Yes

Reviewer #2: Yes

4. Have the authors described where all data underlying the findings will be made available when the study is complete?

Reviewer #1: Yes

Reviewer #2: Yes

5. Is the manuscript presented in an intelligible fashion and written in standard English?

Reviewer #1: Yes

Reviewer #2: No

6. Review Comments to the Author

You may also provide optional suggestions and comments to authors that they might find helpful in planning their study.

Reviewer #1: The protocol of this systematic review aims to explore the etiologies of Attention Deficit Hyperactivity Disorder (ADHD), which holds significant clinical importance and research value. The overall structure of the protocol is clear, and the methods are relatively detailed. However, there are still some aspects that need improvement.

1. The retrieval strategies for each database should be elaborated in detail, rather than just providing an example of the search strategy for PubMed. This can ensure the repeatability of the retrieval process and also facilitate other researchers to evaluate the rationality of the retrieval.

2. Please supplement the Sensitivity analysis. Regarding the issues of loss of original research information and subjective influence, explain how to reduce their impact and improve the reliability of research results through rigorous methodologies (such as adopting standardized data extraction and quality assessment methods, conducting sensitivity analysis, etc.) and quality assessment (such as evaluating the background and conflict of interest of the authors of systematic reviews and meta - analyses, and testing the stability of research results, etc.).

3. Refine the classification operation of the health ecological model: Elaborate on the specific factors included in each category (macro - environment, interpersonal relationships, individual congenital characteristics, behavioral lifestyles, and socioeconomic status) in the health ecological model, and provide specific operation guidelines for classification. For example, in the macro - environment category, clarify the definition criteria of urban or rural environments (such as population density, infrastructure, etc.). At the same time, explain how to ensure the accuracy and consistency of classification during the data extraction and analysis process.

4. Please supplement the publication bias analysis.

Reviewer #2: Dear Authors,

Thank you for the opportunity to review this protocol for an umbrella review on the etiology of ADHD. The manuscript presents a well-structured protocol with several strengths, including comprehensive database coverage, clear adherence to established guidelines, and appropriate methodological frameworks. The topic is highly relevant and the proposed review has the potential to provide valuable insights into ADHD etiology.

However, I have some concerns that should be addressed to enhance the protocol's methodological rigor.

A critical point relates to the time frame restriction. The exclusion of systematic reviews and meta-analyses published before January 2019 appears arbitrary and needs justification. While this recent cut-off would likely prevent overlap among included reviews, it might exclude valuable high-quality evidence from older, more focused reviews. The authors should either provide a clear rationale for this restriction or consider expanding the time frame. In the latter case, they would need to address how overlapping studies will be managed, for instance using the corrected covered area index or the meta-umbrella package (https://mentalhealth.bmj.com/content/26/1/e300534).

There is also some confusion regarding the quality assessment methodology. The protocol states that GRADEpro GDT will be used to evaluate the quality of included studies. However, GRADEpro is designed to assess the certainty of evidence for specific outcomes within systematic reviews, not the methodological quality of the reviews themselves. The authors should clarify how they plan to use GRADEpro for evaluating certainty of evidence for specific outcomes, while specifying which tools will be used to assess the methodological quality of the included reviews.

Furthermore, there is ambiguity in the analysis plan. The statement suggesting that a new systematic review or meta-analysis will be conducted on risk factors ("We will do a systematic or meta-analysis on at least one clearly established risk factor...") appears inconsistent with the umbrella review methodology described elsewhere in the protocol. This needs to be clarified to avoid confusion about the intended analytical approach.

Finally, the manuscript would benefit from English language editing. For instance, several sentences inappropriately begin with "And", and there are some typographical errors (e.g., "PROSERP" instead of "PROSPERO").

These revisions would strengthen what is already a promising protocol. I look forward to seeing the final version and eventually the results of this important work.

Best regards,

Antonio Di Francesco

7. PLOS authors have the option to publish the peer review history of their article (what does this mean?). If published, this will include your full peer review and any attached files.

Reviewer #1: No

Reviewer #2: **Yes: **Antonio Di Francesco

---

## [Author Response · Author response to Decision Letter 0]

23 Dec 2024

Really appreciate your very helpful and kind advice. Thank you so much.

---

## [Editor Report · Decision Letter 1]

12 Jan 2025

The Etiology of Attention Deficit Disorder with Hyperactivity: A systematic review protocol

PONE-D-24-44751R1

Dear Dr. Yanru,

We’re pleased to inform you that your manuscript has been judged scientifically suitable for publication and will be formally accepted for publication once it meets all outstanding technical requirements.

Kind regards,

Carmen Concerto

Academic Editor

PLOS ONE

---

## [Editor Report · Acceptance letter]

16 Jan 2025

PONE-D-24-44751R1 

PLOS ONE

Dear Dr. Wang, 

I'm pleased to inform you that your manuscript has been deemed suitable for publication in PLOS ONE. Congratulations! Your manuscript is now being handed over to our production team.

Kind regards, 

on behalf of

Dr. Carmen Concerto 

Academic Editor

PLOS ONE